# Characterization of Nutritional Quality Traits of a Common Bean Germplasm Collection

**DOI:** 10.3390/foods10071572

**Published:** 2021-07-06

**Authors:** Ester Murube, Romina Beleggia, Deborah Pacetti, Ancuta Nartea, Giulia Frascarelli, Giovanna Lanzavecchia, Elisa Bellucci, Laura Nanni, Tania Gioia, Ugo Marciello, Stefania Esposito, Giacomo Foresi, Giuseppina Logozzo, Giuseppe Natale Frega, Elena Bitocchi, Roberto Papa

**Affiliations:** 1Department of Agricultural, Food and Environmental Sciences, Marche Polytechnic University, 60131 Ancona, Italy; e.m.murube@univpm.it (E.M.); d.pacetti@staff.univpm.it (D.P.); a.nartea@pm.univpm.it (A.N.); g.frascarelli@pm.univpm.it (G.F.); g.lanzavecchia@staff.univpm.it (G.L.); e.bellucci@staff.univpm.it (E.B.); l.nanni@staff.univpm.it (L.N.); stefaniaesposito5@gmail.com (S.E.); giacomo.f_91@hotmail.com (G.F.); n.g.frega@univpm.it (G.N.F.); r.papa@univpm.it (R.P.); 2Cereal and Industrial Crops Research Centre (CREA-CI), Council for Agricultural Research and Economics, 71122 Foggia, Italy; romina.beleggia@crea.gov.it (R.B.); ugo.marciello@gmail.com (U.M.); 3School of Agricultural, Forestry, Food and Environmental Sciences, University of Basilicata, 85100 Potenza, Italy; tania.gioia@unibas.it (T.G.); giuseppina.logozzo@unibas.it (G.L.)

**Keywords:** biofortification, plant breeding, plant genetic resources, legumes, nutritional quality

## Abstract

Food legumes are at the crossroads of many societal challenges that involve agriculture, such as climate change and food sustainability and security. In this context, pulses have a crucial role in the development of plant-based diets, as they represent a very good source of nutritional components and improve soil fertility, such as by nitrogen fixation through symbiosis with rhizobia. The main contribution to promotion of food legumes in agroecosystems will come from plant breeding, which is guaranteed by the availability of well-characterized genetic resources. Here, we analyze seeds of 25 American and European common bean purified accessions (i.e., lines of single seed descent) for different morphological and compositional quality traits. Significant differences among the accessions and superior genotypes for important nutritional traits are identified, with some lines showing extreme values for more than one trait. Heritability estimates indicate the importance of considering the effects of environmental growth conditions on seed compositional traits. They suggest the need for more phenotypic characterization in different environments over different years to better characterize combined effects of environment and genotype on nutritional trait variations. Finally, adaptation following the introduction and spread of common bean in Europe seems to have affected its nutritional profile. This finding further suggests the relevance of evolutionary studies to guide breeders in the choice of plant genetic resources.

## 1. Introduction

Common bean (*Phaseolus vulgaris* L.) is the most grown food legume crop worldwide. It is largely grown as a grain crop (i.e., dried beans) and as a fresh vegetable (e.g., snap beans, green beans). In 2018, 30.4 million tons of dry seeds and 24.7 million tons of fresh pods were produced worldwide, with a total of 34.5 and 1.6 million hectares destinated for its cultivation, respectively [1].

From a nutritional point of view, as for legumes in general, common bean represents an important source of protein, complex carbohydrates, dietary fiber, vitamins and minerals, and it has a high energetic value [2,3]. Beans are an excellent source of the dietary protein that has an important role in human nutrition [2]. The most abundant proteins in common bean seeds are the storage proteins, as represented by the globulins, phaseolins (7S fraction, vicilin family) and legumin (11S fraction), and by albumins of the family of lectin and lectin-related proteins (for review, see Sparvoli et al. [4]). Globulins are the major fraction of common bean protein, at up to 50% of total protein [5].

The nutritional importance of legumes is also helped by their low lipid content, at ~2%, depending on the bean variety. They can contain valuable exogenic unsaturated fatty acids, as mainly palmitic, oleic, and linoleic acids (for review, see Hayat et al. [3]). The major lipid components in beans are phospholipids and triacylglycerols, with traces of minor components, such as diacylglycerols. In addition to high protein content and low lipid content, the nutritional value of common bean is linked to its minor compounds, such as vitamins and minerals [2,3,6,7,8]. Beans have the highest mineral content of crop legumes [9] and have been described as an important source of inorganic minerals, especially for iron, zinc, copper, phosphorous, and aluminum [2,10]. Moreover, the biological function of the numerous bioactive compounds that characterize common bean has been related to prevention and/or regulation of chronic diseases, such as obesity, diabetes, cancers, and coronary heart disease (for review, see Messina [11]).

Genetic improvement is one of the most viable strategies to obtain varieties that in addition to having improved agronomic traits, such as yield and resistance to biotic and abiotic stress, also have the best nutritional characteristics to satisfying consumer needs [12]. Exploiting the diversity in plant genetic resources represents a very efficient strategy to increase the nutritional quality of elite varieties. However, to do this, characterization of the plant resources is needed. For exploration and use of the genetic variation in plant genetic resources, there is the need to know the evolutionary history of the crops that shaped the current levels and structure of genetic diversity of the germplasm (see Bitocchi et al. [13]). In this regard, *P. vulgaris* has its origins in Mesoamerica [14], from where wild forms dispersed into South America, which promoted the emergence of three eco-geographically and genetically differentiated wild gene pools: Mesoamerican, Northern Peru and Ecuador, and Andean. Domestication of the species was subsequent to this separation, and occurred independently in the two main gene pools, Mesoamerican and Andean (for review, see Bitocchi et al. [15]).

*Phaseolus vulgaris* was introduced into Europe in the 16th century, after the discovery of the Americas. It then spread throughout Europe, following complex routes that involved multiple exchanges between different European countries, with continued introduction of new varieties from the Americas [16]. Although materials from both the Mesoamerican and Andean gene pools were introduced, prevalence of Andean types has been described within the European germplasm [17,18,19,20,21]. In contrast to the Americas, in Europe, the absence of geographic barriers among the two gene pools promoted inter-gene-pool hybridization. This led to the generation of great genotypic diversity that not only mitigated the effects of the bottleneck that occurred when common bean was introduced into Europe, but also favored establishment of new combinations of traits, to enhance the adaptation to new environments (i.e., resistance/tolerance to biotic, abiotic stresses) [21].

Seed compositional traits in common bean genetic resources have been evaluated in the literature [22,23,24,25,26,27,28,29,30,31,32,33,34]. However, most of these studies were not based on materials grown in one or more specific field trials that involve replicates. Evaluation of genetic resources for seed quality traits needs to be carried out by using appropriate replicated field trials and, possibly, conducted in more than one location and/or year. This allows to obtain reliable estimates of variance components and trait heritabilities. This aspect is particularly important, especially because quality traits are highly influenced by environmental conditions.

In the present study, we aimed to characterize and describe nutritional quality traits of 25 different American and European domesticated genotypes of common bean to identify potential sources of improved seed quality traits that can be used in future breeding programs. We also investigated whether the process of adaptation to European environments had any effects on seed compositional traits.

## 2. Materials and Methods

A total of 25 common bean domesticated accessions were used in this study (Table 1), 13 from America and 12 from Europe. Most are landraces (20 lines), while the remaining are cultivars (5 lines). These accessions represent a subsample of the materials used in the BEAN_ADAPT project, which is an international collaborative project funded through the second ERA-CAPS call, ERA-NET for Coordinating Action in Plant Sciences. The BEAN_ADAPT project is based on nested core collections of different sizes (*Pv_all*, *Pv_core1*, *Pv_Core2*) that were developed to be representative of both American and European germplasm, and of the two main gene pools of the species: Mesoamerican and Andean.

All of the materials are lines of single seed descent that were purified by one or more cycles of selfing under insect-free conditions. This allows association of phenotypes to reliable genotypic information and represents strong potential to promote genetic resources used in pre-breeding and breeding programs. *Pv_all*, *Pv_core1*, and *Pv_core2* comprise ~10,000, 500, and 200 domesticated lines of *P. vulgaris*, respectively. The 25 accessions included in the present study were part of *Pv_core1*. Within the activities planned for the BEAN_ADAPT project, genotyping-by-sequencing of *Pv_core1* provided the data for the population structure analysis that highlighted the main structure as two genetic clusters, which represent the Mesoamerican and Andean gene pools (Roberto Papa, personal communication).

In the present study, with a threshold of memberships (*q*) >50% used to assign the genotypes to one of the two gene pools, 20 were Mesoamerican, and five were Andean (Table 1). Fifteen accessions were completely assigned (*q*, 100%) to the Mesoamerican gene pool, while five Mesoamerican accessions showed introgression from the Andean gene pool (Mesoamerican gene pool *q*, 0.68–0.94). One accession was completely assigned to the Andean gene pool (*q*, 100%), while the remaining four Andean accessions showed introgression from the Mesoamerican gene pool (Mesoamerican gene pool *q*, 0.04–0.33).

The morphology and nutritional quality traits of these 25 accessions were analyzed using seeds produced during a field experiment involving all of the 500 *Pv_core1* BEAN_ADAPT lines carried out in southern Italy in 2017, at the experimental farm of the *Istituto Tecnico Agrario ‘Rocco Scotellaro’* (Marsicovetere, Potenza, Italy; latitude, 40°19′56.7″ N; longitude, 15°49′28.4″ E; altitude, 592 m a.s.l.). This is an internal intensive horticultural area, characterized by a temperate Mediterranean climate with cool winters and warm dry summers. The data of maximum and minimum temperature and rainfall for the growing season (May to October) were obtained from the nearest weather station located, as shown in Figure 1. The soil at the location is a fertile coarse lime soil. The field experiment was conducted as a randomized complete block design with four replicates. Seeds were sown on 22 June 2017, in single-row plots with nine seeds per plot (1.2 m between rows; 0.2 m between plants; 0.9 m between plots). Triple superphosphate (TSP, 46% P_2_O_5_) fertilizer was applied as basal fertilization before sowing. Standard conventional agronomic management and irrigation practices were applied to the experimental fields. Plastic cover was used for weed control. Pest and disease management was done by spraying the plants with Ridomil Gold (fungicide) and Klartan 20 Ew (against aphids).

### 2.1. Morphological Seed Traits

Ten pods for each plot and replicate were harvested at fully maturity (R9, when 90% of pods on the plant were golden-brown) and threshed. Seed weight (mean seed weight of all the harvested seeds), cotyledon and embryo weights, and seed coat weight and thickness were determined. For cotyledon, embryo, and seed coat weights, seeds were soaked in distilled water for 24 h. When the seeds were fully hydrated, the cotyledons, seed coat, and embryo were removed and dried in silica gel. When the components of the seeds were completely dry, the weights of the cotyledons, embryos and seed coat were determined using a precision scale, and the thickness of the seed coat was measured with a digital micrometer (mm; Mitutoyo 164–163). Cotyledon, embryo, and seed coat weights for each accession are reported as mg/g seeds.

### 2.2. Seed Nutritional Quality Traits

The seeds were lyophilized and milled to a fine powder using a planetary mill with an agate jar and balls (Pulverisette 7 Planetary Micro 200 Mill, Classic Line; Fritsch GmbH Milling and Sizing, Idar-Oberstein, Germany). The following seed nutritional traits were measured.

#### 2.2.1. Protein Content

Protein content was determined using Dumas combustion. The N content was measured using a N/protein analyzer (Leco FP-528; LECO Corp., St Joseph, MI, USA), as described in the Official Methods of Analysis, Dumas method, 990.03 [35].

#### 2.2.2. Tocopherol Analysis

For the tocopherol analysis, 5 g to 10 g of powdered seeds obtained by lyophilizing and grinding was dissolved in chloroform:methanol (2:1, *v*/*v*) for extraction of the lipid matter, at room temperature and with magnetic stirring. After 3 h, the content was divided in two tubes and centrifuged at 5000× *g* for 15 min. The lipid-free powder pellet was discarded. The solvent was transferred to a new tube and removed in a rotary evaporator (Rotavapor R-300, Buchi, Italy). The remaining oil was recovered with acetonitrile and transferred into a 2 mL vial. The total recovered lipids for each accession was weighed. The lipid content was expressed as g/100 g of dry seed.

The different tocopherol components were identified according to Giardinieri et al. [36]. Briefly, 10 µL of the extracted oil was analyzed using ultra-high-performance liquid chromatography coupled to a fluorescence detector. The column (Express C18 column; 7.5 cm × 2.1 mm, 2.7 μm; Ascentis, Supelco) and autosampler temperatures were set at 25 °C. The injection volume was 1 µL, and the mobile phase was acetonitrile/methanol (90:10, *v*/*v*), at a flow rate of 0.45 mL/min. Fluorescence detection was at 296 nm excitation and 325 nm emission. For quantification of tocopherols, seven standard stock solutions of γ-tocopherol in acetonitrile were prepared and analyzed (5–150 µg/mL; correlation coefficient, R = 0.997).

#### 2.2.3. Antioxidant Activity

The antioxidant activity was measured according to the α, α-diphenyl-β-picrylhydrazyl (DPPH) method [37]. The Trolox equivalent antioxidant activity (TEAC) was calculated for each sample, as compared to the Trolox standard. For this, 10 (±0.5) mg of each sample was incubated with 10 mL of DPPH solution (5 mg/100 mL) under stirring in the dark for 30 min. After brief centrifugation, absorbance at 525 nm was measured. For the calibration curve, five standards were prepared with Trolox ranging from 0 µM to 30 µM.

The percentage of inhibition for the samples and Trolox was calculated according to Equation (1):(1)$$\%\;\mathrm{Inhibition}\;\left(\mathrm{sample}\;\mathrm{or}\;\mathrm{Trolox}\right)=\frac{\left(\mathrm{Abs}\;\mathrm{blank}\;–\;\mathrm{Abs}\;\mathrm{sample}\;\mathrm{or}\;\mathrm{Trolox}\right)}{\mathrm{Abs}\;\mathrm{blank}}\times 100$$

The TEAC of each sample was as given by Equation (2):(2)$$\mathrm{TEAC}\;\left(\mathrm{mmol}\;\mathrm{equivalent}\;\mathrm{Trolox}/\mathrm{kg}\;\mathrm{sample}\;\mathrm{DW}\right)=\%\;\left(\mathrm{inhibition}\;\mathrm{sample}\;/s\times m\right)\times 10$$
where s is the calibration curve slope, m is the weight in mg, and 10 is the dilution factor.

#### 2.2.4. Yellow Pigment Content

Total carotenoid content was reported as yellow pigment content (YPC), according to the micro-method described by Beleggia et al. [38,39] and expressed as mg per kg on dry matter of β-carotene. Briefly, 50 mg of the seed powder of each sample was used in the extraction with water-saturated 1-butanol. After brief centrifugation, absorbance at 435 nm was measured with a UVvis HPLC detector (UV-900; Amersham Pharmacia Biotech, Uppsala, Sweden). YPC was calculated according to Equation (3):(3)$$\mathrm{YPC}\;(\frac{\mathrm{mg}}{\mathrm{kg}})=\left(\left(\mathrm{ABS}435\mathrm{nm}/1.6632\right)\times 2.5/\mathrm{sample}\;\mathrm{weight}\;\left(\mathrm{kg}\right)\right)$$
where 1.6632 is the absorbance at 435 nm of 1 mg β-carotene in 100 mL solvent (water-saturated 1-butanol), and 2.5 is the correction factor.

### 2.3. Mineral Content

To determine the mineral contents (i.e., macro-elements: calcium, Ca; potassium, K; magnesium, Mg; sodium, Na; phosphorous, P; micro-elements: copper, Cu; iron, Fe; manganese, Mn; molybdenum, Mo; nickel, Ni; selenium, Se; zinc, Zn), 20 mg of each sample was dissolved in 8 mL HNO_3_ (69.5%)/2 mL H_2_O_2_ (30%) using a microwave oven (Ethos Touch Control; Milestone, Sorisole, Italy), and then diluted to 50 mL with high purity deionized water acidified with 3% HNO_3_. The mineral contents were then determined using inductively coupled plasma mass spectrometry (7700×; Agilent Technologies, Italy) with an auto-sampler (ASX-500; Agilent Technologies, Italy), as described by Beleggia et al. [40]. Briefly, the inductively coupled plasma mass spectrometry was tuned to standard mode and with collision gas (He), to remove many of the simple solvents and the argon-based polyatomic spectral interference. The plasma power was operated at 1550 ± 50 W, and the carrier and make-up gases were typically set at 0.83 L/min and 0.17 L/min, respectively. Sample uptake was maintained at ~0.1 mL/min using a self-aspirating nebulizer. Data were processed using the MassHunter WorkStation software (Agilent Technologies, Italy), and the mineral contents were quantified using external calibration curves and are expressed as g/kg for macro-elements and mg/kg for micro-elements. The microwave digestion program settings were defined as follow: stage 1, max power 600 W, ramp 3 min, temperature 120 °C, pressure 350 PSI, time 5 min; stage 2, max power 600 W, ramp 15 min, temperature 200 °C, pressure 350 PSI, time 10 min.

### 2.4. Data Analysis

Data for the following 23 traits were used for the analysis (Appendix A): (i) morphological seed traits: average seed weight (g), cotyledon, embryo, and seed coat weights (mg/g seeds), seed coat thickness (mm); (ii) nutritional quality traits: lipids (g/100 g lyophilized), γ-tocopherol (mg/100 g of lipids), δ-tocopherol (mg/100 g of lipids), antioxidant activity (TEAC; mmol Trolox/kg dry seed), yellow pigment content (YPC; mg β-carotene/kg dry seed), and protein content (%); (iii) macro-element contents (Ca, K, Mg, Na, P; g/kg); and (iv) micro-element contents (Cu, Fe, Mn, Mo, Ni, Se, Zn; mg/kg). The evaluation of such traits was done in seed samples obtained in a proper replicated experimental field trial, thus for each trait and line four evaluations were available, allowing to accurately estimate the different parameters.

The statistical analysis was performed using RStudio v1.2.5001 [41], with the custom scripts. For all of the traits considered, mean and standard deviation for each accession were computed. Associations between variables were quantified using Pearson’s correlation method and pairwise complete observations, with the package *corrplot* [42].

Differences among accessions for all of the quality traits considered were tested using one-way analysis of variance (ANOVA). Mean discrimination was performed, by applying Tukey’s tests, and statistically significant differences were determined at the probability level of *p* < 0.05, using the R package *agricolae* v 1.3-3 [43]. To calculate the variance components, estimate from ANOVA analysis, a random model (restricted maximum likelihood method) was used to highlight the contribution of each variance component: accession, replicate, and replicate within accession. For this, the package *minque* v 1.1 [44] was used. Heritability (h^2^) was computed as the ratio between genotypic variance and phenotypic variance [45]. Wilcoxon–Kruskal–Wallis nonparametric test [46] was used to test for differences in antioxidant activities between accessions with white and colored seeds.

The differences between the American and European accessions were tested by an analysis of covariance (ANCOVA), including population structure information as covariate in the model. The multivariate analysis, principal component analysis (PCA), was performed with the packages *FactoMiner* v1.42 [47] and *factoextra* v1.0.5 [48]. PCA was initially computed based on morphological seed traits, nutritional quality traits, and total macro-elements and micro-elements. A second PCA was computed based on only the macro-elements and micro-elements per accession. All of the figures were constructed using the *ggplot2* v3.2.1 package [49].

## 3. Results

### 3.1. Variability among Accessions

Means (±standard deviation) were computed for all of the traits analyzed in each accession (Appendix A), with the ranges shown in Table 2.

Variability among the different genotypes was seen for all of the traits considered (Table 2 and Appendix A). ANOVA analyses highlighted significant differences between the accessions for all of the morphological seed traits, except for seed coat thickness (Table 2; Figure 2 and Appendix A). Seed weights ranged from 0.2 g to 0.6 g (Table 2), with European accessions ECe095 and ECe171 showing the highest seed weights (Figure 2; Appendix A).

Variation was also detected for the different parts of the seeds, as cotyledon, embryo and seed coat weights, which ranged from 879.1 mg/g to 924.3 mg/g, from 9.1 mg/g to 16.1 mg/g, and from 62.7 mg/g to 106.2 mg/g, respectively (Table 2). In particular, the American accession ECa103 showed the highest value for cotyledon weight and the lowest for seed coat weight. The opposite was observed for the American accession ECa078 that showed the lowest and highest values for cotyledon weight and seed coat weight, respectively (Appendix A). American accessions ECa052 and ECa203 and European accession ECe177 showed the highest embryo weights, with the lowest for European accession ECe166 (Appendix A).

Even though wide ranges were seen for lipid content (1.8–3.1 g/100 g of dry seed), γ-tocopherol (43.3–127.2 mg/100 g of lipids), and the ratio between γ-tocopherol and δ-tocopherol (7.7–19.8), significant differences among the accessions were only seen for δ-tocopherol (Table 2; Appendix A). Here, the European accessions ECe092 and ECe234 showed the highest (13.6 mg/100 g of lipids) and lowest (3.7 mg/100 g of lipids) δ-tocopherol contents, respectively (Appendix A).

High variability was seen for antioxidant activities and carotenoids contents (Table 2; Appendix A). The accessions showed significantly different values for these quality traits (Figure 2); in particular, American accession ECa200 showed the highest antioxidant activity (TEAC, 26.2 mmol Trolox/kg dry seed), while a group of accessions mainly from Europe (ECa046, ECe312, ECa102, ECe177, ECe092, ECe166, ECe294, ECe095) showed very low antioxidant activities (≤7.1 mmol Trolox/kg dry seed) (Figure 3a; Appendix A). Accessions with colored seeds showed higher antioxidant activities compared to those with white seeds (Wilcoxon signed-rank test, *p* = 0.001). The carotenoids contents ranged from 2.2 to 9.3 mg β-carotene/kg dry seed (Table 2), and was highest for American accession ECa060 (YPC, 9.3 mg β-carotene/kg dry seed) and lowest for ECe166 (YPC, 2.2 mg β-carotene/kg dry seed) (Figure 3b; Appendix A). Significant differences among the accessions were seen for protein contents (Table 2), with European accession ECe092 as the highest (24.8%) (Appendix A).

For macro-elements and micro-elements, wide ranges were observed (Table 2 and Appendix A). Among the macro-elements, ANOVA F-test was significant for Na, P and Ca, however significant differences among the accessions (Tukey tests) were only seen for Na and Ca, with European accession ECe122 showing the highest content for both of these macro-elements, at 0.5 g/kg and 4.6 g/kg, respectively (Table 2; Figure 4a and Appendix A). For three accessions from America (ECa060, ECa078, ECa095), no Na was detected in the seeds (Appendix A). All of the micro-elements showed significant differences among the accessions (Table 2; Figure 4 and Appendix A). The European accession ECe122 showed the highest micro-elements content (252.8 mg/kg) (Appendix A). Cu content ranged from 7.0 mg/kg to 16.5 mg/kg (Table 2), where European accession ECe092 showed the highest Cu content (16.5 mg/kg), with the lowest for American accession ECa046 (7.0 mg/kg) (Appendix A). Fe content ranged from 55.4 mg/kg to 176.1 mg/kg (Table 2), with European accession ECe122 showing the highest Fe content (176.1 mg/kg) (Appendix A). Mn ranged from 15.9 mg/kg to 34.8 mg/kg (Table 2), with the highest and lowest Mn in accessions ECa052 and ECe171, respectively (Appendix A). The highest (77.7 mg/kg) and lowest (5.1 mg/kg) Zn contents were seen for American accession ECa046 and European accession ECe268, respectively. The highest Mo was seen for European accession ECe234 (4.3 mg/kg), and the lowest for European accession ECe095 (0.7 mg/kg) (Appendix A). The two remaining micro-elements showed significant differences among the accessions, as Se and Ni (Table 2). Se ranged from 0.0 mg/kg to 4.1 mg/kg; interestingly for 16 of the 25 accessions (64%) no Se was detected in the seeds (Figure 4b; Appendix A). Of these, eight were American (ECa046, ECa052, ECa060, ECa064, ECa078, ECa085, ECa102, ECa200) and eight were European (ECe017, ECe092, ECe095, ECe166, ECe171, ECe177, ECe234, ECe240); the ECa096 accession showed the highest Se (4.1 mg/kg) (Figure 4b; Appendix A). For only one accession (ECe240) Ni was not detected, while the highest Ni was detected in ECa095 (4.9 mg/kg) (Appendix A).

### 3.2. Heritability

Heritability was estimated for all the traits analyzed (Table 2 and Appendix A). The traits with the highest heritability estimates (h^2^ > 60%) were seed weight, antioxidant activity, carotenoids content, Na, Mo, and Se contents. Intermediate heritabilities (0.30 < h^2^ < 0.60) were detected for embryo, cotyledon and seed coat weights, δ-tocopherol, protein, and Ca, Cu, Fe, Mn, Ni, and Zn contents. Very low heritability estimates (h^2^ < 0.30) were seen for the remaining traits, seed coat thickness, lipids, γ-tocopherol, and Mg, P and K contents.

Considering the morphological seed traits, the highest h^2^ estimate was for seed weight (h^2^, 91.4%), followed by embryo weight (h^2^, 52.7%); seed coat and cotyledon weights, and seed coat thickness showed low h^2^ estimates (h^2^, 39.6%, 36,6%, 10.2%, respectively). Among the nutritional quality traits, high h^2^ estimates were seen for antioxidant activity and carotenoid content (h^2^, 84.1%, 69.4%, respectively). The heritabilities for δ-tocopherol and protein content were 31.5% and 30.5%, respectively. Very low h^2^ estimates were detected for γ-tocopherol (h^2^, 12.7%) and lipid content (h^2^, 0.0%), revealing the strong environmental influences on these traits and/or no differences within these specific materials. Finally, high h^2^ was seen for Na, Se, Mo, Zn, and Ni (h^2^, 84.5%, 81.2%, 67.3%, 53.3%, 52.2%, respectively). The Ca macro-element and the Mn, Cu and Fe micro-elements showed moderate heritability estimates (h^2^, 35.9%, 38.9%, 35.3%, 39.5% respectively). Low heritability estimates were seen for the remaining elements (h^2^ ≤ 22.4%).

### 3.3. Differences between American and European Germplasm

ANCOVA analysis was carried out to test for significant differences among the American and European accessions for the morphological and compositional traits of the seeds. For the morphological traits, compared to the American accessions, the European accessions showed significantly higher seed weights and lower embryo weights (Table 3; Figure 2 and Appendix A). For the traits related to the lipids in the seeds, significant difference between the American and European accessions was only seen for δ-tocopherol, with those European showing higher δ-tocopherol than those American (Table 4; Appendix A).

Strong differences were detected for the antioxidant activities and carotenoids content. In particular, these were both higher for the American accessions (Table 4; Figure 3). The TEAC of the American and European accessions was 17.2 mmol and 10.3 mmol Trolox/kg dry seed, respectively. For YPC, 5.6 and 4.0 mg β-carotene/kg dry seed were detected for the American and European accessions (Table 4; Figure 3). The European accessions showed slightly, but significantly, higher protein content compared to the American ones (Table 4; Appendix A).

American and European accessions differed significantly for three macro-elements. In particular, compared to those American, the European accessions showed higher Na and Ca contents, and lower Mg content (Table 5; Appendix A). Within the micro-elements, significant difference was only seen for Se. For the rest of the macro-elements and micro-elements analyzed, there were no significant differences seen (Table 5).

### 3.4. Principal Component Analysis

Principal component analysis was performed to investigate the relationships among the accessions, and to determine which seed traits define the basis of their differentiation. Considering PCA analysis based on all of the morphological traits, nutritional traits, and the macro-elements and micro-elements (Figure 5), principal component 1 (PC1) and PC2 explained 26.5% and 16.1% of the total variance, respectively.

Here, PC1 was significantly and positively correlated with seed weight, γ-tocopherol, cotyledon weight, δ-tocopherol, and protein content, and negatively correlated with seed coat weight. embryo weight, YPC, and TEAC (Table 6), as also highlighted by the loading plot from the PCA analysis showing that seed weight and δ-tocopherol contribute more for clustering the accessions (Figure 5).

PC1 tends to separate the American and European accessions, whereby those from Europe have higher mean seed weight and protein content, and higher mean content of both tocopherol congeners, while the American accessions showed higher TEAC and YPC activities (Figure 5), in agreement with the data from ANCOVA (Table 4 and Table 5). PC2 was significantly and positively correlated with the δ-tocopherol, lipid content and embryo weight, while it was negatively correlated with cotyledon weight and ϒ/δ ratio (Table 6), which contributes more to cluster accessions as also highlighted by the loading plot (Figure 5). Interestingly, the European accessions were more dispersed than the American ones, which suggested wider variability for the seed traits considered here.

Figure 6 shows the PCA analysis based on only the macro-elements and micro-elements, where PC1 and PC2 explained 23.7% and 18.0% of the total variation, respectively. PC1 was significantly and positively correlated with Ca, Fe, Mn, Na and Se contents and negatively correlated with P content, while PC2 was significantly and positively correlated with Mo, Mg, K and P (Table 7), in agreement with the lodging plot of the PCA analysis (Figure 6).

No clear distinction was detected in these analyses among the accessions, even if the macro-elements and micro-elements also showed wider variability for the European accessions (Figure 6). European accessions ECe234 and ECe122 were separated from the other ones; in particular, ECe234 showed higher contents of K, and Mo, and among the highest contents of Mg and Fe, while accession ECe122 showed high contents of Fe, Na and Ca, and among the highest contents of Mn and Se (Figure 4, Figure 6, Appendix A).

### 3.5. Correlations between the Traits

Correlations between traits were also investigated (Figure 7; Appendix A). For morphological seed traits, seed weight was significantly and positively correlated with seed coat thickness (r = 0.21, *p* < 0.05), and negatively correlated with embryo weight (r = −0.47, *p* < 0.01). A strong negative and significant correlation was seen between cotyledon and both seed coat weight and embryo weights (r = −0.98, −0.58, respectively; *p* < 0.001). For lipids and γ-tocopherol and δ-tocopherol contents, these were significantly and positively correlated (r = 0.73, *p* < 0.001), while γ-tocopherol content was negatively correlated with lipid content (r = −0.30, *p* < 0.01) and δ-tocopherol was positively correlated with protein content (r = 0.26, *p* < 0.01), but negative correlated with TEAC (r = −0.30, *p* < 0.01) and YPC (r = −0.26, *p* < 0.01). TEAC and YPC were positively correlated (r = 0.5, *p* < 0.001). Significant correlations were also seen between the morphological and nutritional traits; in particular, seed weight was positively correlated with γ-tocopherol content (r = 0.21; *p* < 0.05) and protein content (r = 0.24, *p* < 0.05), and strongly negatively correlated with TEAC and YPC (r = −0.38, −0.47, *p* < 0.001). TEAC was also positively correlated with embryo weight (r = 0.29, *p* < 0.01) and seed coat weight (r = 0.23, *p* < 0.05) and negatively correlated with cotyledon weight (r = −0.26, *p* < 0.01).

All of the significant correlations between the different macro-elements and micro-elements were positive, except for the macro-element P, which was negatively correlated with Na, Ca, Mn, and Se (r = −0.26, −0.21, −0.23, −0.27, respectively; *p* < 0.05). Among the macro-elements and micro-elements, the positive correlations that showed significance (*p* < 0.001) were those between Ca and Na (r = 0.34), Mn and K (r = 0.43), Mn and Ca (r = 0.41), Mg and Mo (r = 0.34), Mg and K (r = 0.64), and Fe with Ca (r = 0.52), Na (r = 0.42) and Mn (r = 0.33). K and Mg were strongly correlated with the total amount of macro-elements (r = 0.88, 0.70, respectively, *p* < 0.001), and Fe content with the total micro-elements content (r = 0.88, *p* < 0.001).

Considering seed weights and macro-elements and micro-elements, significant positive correlations were detected between seed weight and Na (r = 0.34, *p* < 0.001), Zn (r = 0.33, *p* < 0.001), and Ni (r = 0.24, *p* < 0.05). Seed weight was significantly negatively correlated to Mn, Se, and Mo (r = −0.30, −0.26, −0.22, respectively; *p* < 0.05). Significant negative correlation was seen between embryo weight and Na content (r = −0.34, *p* < 0.001); cotyledon weight and Mg content were negatively correlated (r = −0.22, *p* < 0.05), while seed coat weight was positively correlated with Mg and Zn (r = 0.23 and 0.21, respectively; *p* < 0.05).

Focusing on the nutritional traits, the macro-elements and micro-elements showed different significant correlations; among the most significant, Na and Zn were negatively correlated with TEAC (r = −0.35, −0.31, respectively; *p* < 0.01) and YPC (r = −0.30, −0.26, respectively; *p* < 0.01). YPC and Mn content were strongly correlated (r = 0.38, *p* < 0.001).

## 4. Discussion

We present here the characterization of 25 highly phenotypically and genotypically differentiated common bean accessions in terms of several seed quality traits. These data contribute toward exploitation of their genetic diversity for improvement of nutritional quality of common bean varieties. The present study has great value that arises from the sample used, to offer additional value for their exploitation in plant breeding, according to the following characteristics: (i) the seeds of the different accessions were derived by single seed descent, and are thus identical at the genome level; (ii) the majority of the accessions were landraces, thus potentially including variability that is not present in modern varieties; (iii) the seeds included accessions that were representative of all of the different growth habit types, as a character that is associated with several agronomic traits and with common bean races [50,51,52,53]; (iv) the sample was balanced for American and European origins, thus making it possible to investigate the potential effects on these morphological and compositional traits of the seeds due to adaptation of the common bean to the European environments, after the introduction and spread from its centers of origin; and (v) the seeds used for morphological and chemical analysis were obtained in a designed experimental field trial (with four replicates), to allow estimation of the variance components and trait heritabilities.

### 4.1. Identification of Nutritionally Enhanced Genotypes

Large variability was seen for all the traits analyzed, with the exception of lipid content, for which the heritability estimate was zero, and for seed coat thickness, γ-tocopherol, and Mg, and K contents, for which no significant differences and low heritabilities were seen. Thus, the discussion focuses mainly on the traits that showed significant differences among the accessions and intermediate-to-high heritability estimates.

### 4.2. Morphological Seed Traits

Seed weight was the trait that showed the greatest estimate of heritability (h^2^, 91.4%) in the present study. This was expected, as the literature is full of studies that have reported such observations for common bean [54,55,56]. This outcome is also consistent with what has been shown for other legume crops, such as chickpea [57,58], lentil [59], faba bean [60], soybean [61] and pea [62].

Significant differences among the 25 accessions were detected for all the morphological seed traits, with the only exception being seed coat thickness, for which no differentiation was detected among these accessions. Accessions ECe095 and ECe171 showed the highest seed weights compared to the other genotypes; both are European landraces. Morphological traits of seeds are important to be considered, not only because the seed size, shape, and color are important end-user features that are valued by consumers, but also because they can be correlated to nutritional and quality traits, such as, e.g., the relationship between seed coat percentage and cooking time [63]. In particular, in the present study we showed significant correlations for: (i) several nutritional traits (i.e., γ-tocopherol, antioxidant activity, YPC, protein, and Na, Zn, Mn, Se, Mo, Ni contents) with seed weight; (ii) four nutritional traits (i.e., lipid content, antioxidant activity, Mg and Zn contents) with seed coat weight; (ii) embryo weight with antioxidant activity and Na content; and finally, (iv) cotyledon weight with antioxidant activity and Mg.

### 4.3. Lipids and Tocopherols

The lipid content for these samples ranged from 1.8 to 3.1 g/100 g of dry seed. This range is comparable to those in other studies based on analysis of seeds from a few common bean cultivars [10,27,64,65,66], and it confirms that common bean is not an oil-bearing legume species. We did not detect significant differences for lipid content among the different genotypes, and the heritability estimate was zero, apparently due to a large influence of environment on the expression of the trait; however, also low genetic variance for the specific trait in the analyzed samples might explain these findings.

Increasing the number of genotypes, and especially including landraces and wild forms, can help to improve predictions of breeding value and identify genotypes with extreme values for lipid content. This was suggested by the study of Chávez-Servia et al. [67] where they analyzed 70 landraces from Mexico, the center of origin of the species. Here they reported a wider range for the lipid content (1.5–6.2%) compared to those present in the literature based on characterization of very small samples of modern cultivars. For the lipids, in the present study, we focused on tocopherols. The tocopherols (i.e., α, β, δ, γ isomers of Vitamin E) are oil-soluble compounds with antioxidant properties, and indeed their main function is to protect biological membranes from peroxidation [68,69,70]. Even if α-tocopherol is considered the main contributor to Vitamin E activity, it has been suggested that γ-tocopherol is a more efficient antioxidant [71]. In pulses, the distributions of these four isomers are different; in particular γ-tocopherol is the most abundant isomer in all legumes, whereas α-tocopherol and δ-tocopherol contents are lower, and these are not seen for all legume species [72,73]. For all of the 25 common bean genotypes analyzed in the present study, only the γ and δ isomers were detected, with the former ~11-fold more abundant than the latter.

This has also been reported in the literature [28,29,72], although there are some studies where α-tocopherol was reported for seeds of some common bean varieties [66,73,74]. Several studies have indicated the beneficial effects of correct dietary intake of Vitamin E, as it prevents free radical propagation reactions, which are the base mechanisms involved in many human diseases, such as cancers and cardiovascular disease [75]. The contents of γ-tocopherol and δ-tocopherol that were detected in these samples were consistent with analyses in the literature; however, to the best of our knowledge, the present study is the first based on analysis of a sample of 25 different common bean genotypes, the seeds of which were obtained in a designed experimental replicated field trial. This allowed us to investigate the phenotypic variance components for these traits, and to make inferences on the breeding value of the materials considered.

The two tocopherol isomers were positively and significantly correlated. Positive correlations between different tocopherol isomers were also reported for chickpea cultivars [76,77]. The more abundant tocopherol (i.e., γ-tocopherol) showed low heritability (12.7%), with no significant differences detected among the genotypes. This can be explained by several findings that have indicated that tocopherol content and composition in plants are strongly affected by the environmental conditions under which the plants are grown, particularly for abiotic stress, such as high-intensity light, drought, salinity, and low temperatures [78,79,80,81]. In contrast, δ-tocopherol varied significantly among genotypes and showed an intermediate heritability estimate (31.5%). European accession ECe092 was a landrace from Greece, and it showed the highest δ-tocopherol content (13.5 mg/100 g of lipids); it can thus be considered very interesting for breeding aimed at increasing the content of these compounds in common bean seeds. Due to the low content of lipids in seeds, the real potential of common bean as source of tocopherols to satisfy the recommended dietary allowance (RDA) is low, however it can contribute to this regard, especially in developing countries, where it is a staple food.

### 4.4. Antioxidant Activity and Carotenoids Content

Several studies have shown that pulses can be valuable sources of natural antioxidants (e.g., Singh et al. [82,83]). The antioxidant activities determined in the present sample varied considerably (1.1–26.2 mmol Trolox/kg dry seed), with this trait showing a high heritability estimate (84.1%). Moreover, significant differences were detected among the genotypes, where accession ECa200, which is a Brazilian landrace, showed the highest antioxidant activity. A lot of studies have focused on characterization of legume seeds in terms of their antioxidant activities, including for common bean (for review, see Singh et al. [82]). Recently, Nadeem et al. [34] evaluated 182 common bean landraces and six commercial cultivars collected from Turkey for their seed antioxidant activities under four environments and in two locations. In agreement with our findings, they showed high heritability for this trait (92%), as well as wide variation among the different landraces, which confirms the great potential for the best performing genotypes to be used successfully in breeding to develop improved varieties with health benefits.

It is well reported in the literature that the antioxidant activity is higher in the seed coat compared to the cotyledons, due to the higher content of phenolic compounds in this part of the seed [28,29,31,82,84,85,86]. These studies have also shown that the colored seeds (i.e., with dark seed coat) are characterized by higher antioxidant activity compared to white beans. Although we did not evaluate the antioxidant activity separately for seed coat and cotyledons, there was positive correlation between antioxidant activity and seed coat weight, in agreement with most of the phenolic compounds being in the seed coat [82]. We also showed that white beans are characterized by very low antioxidant activity compared to colored beans, which suggests that the use of genotypes with pigmented seeds in breeding will be advantageous to enrich common bean seeds in antioxidant activity.

The content of total carotenoids (reported as YPC) was also determined here. They are important in human nutrition as a precursor of Vitamin A, as well as antioxidant, with the demonstration that carotenoid-rich diets are associated with decreased incidence of diverse diseases [87]. Interestingly, for carotenoids content, we found high heritability (69.4%) and significant differences among the different genotypes. The Mexican landrace ECa060 showed the highest YPC, along with high antioxidant activity. The Mexican cultivar ECa095 is also interesting, as it was characterized by high values for both of these traits. Phenotypic evaluation and relative data for carotenoids content in dry common bean seeds are lacking in the literature. Chen et al. [28] analyzed seven cultivars of cranberry beans for carotenoids content and detected only β-carotene at a level comparable to the present findings.

Antioxidant activity significantly and positively correlated with YPC. A similar result was reported by [88], who analyzed carotenoids contents in six commercial high-yield corn hybrids, and identified lutein and β-carotene as the primary contributors to TEAC activity.

The total tocopherol content was not correlated with either antioxidant activity or YPC, which were instead negatively and significantly correlated with δ-tocopherol. Significant correlation between tocopherols and antioxidant activity has been shown in other species, which suggests that the phenotypic architecture of the antioxidant activity might be highly variable among species. Zhang et al. [70] reported weak but significant correlation between tocopherols and DPPH activity across 20 Canadian lentil cultivars. As they also reported increased correlation with DPPH after combining the contents of carotenoids and tocopherols, this suggested that both of these contribute to the DPPH scavenging activity due to a synergistic effect. Lee et al. [77] evaluated 28 cultivars of soybean for tocopherols content and antioxidant activity, and they showed that the γ-tocopherol and δ-tocopherol contents in soybean seeds were strongly and positively related to the antioxidant activity. However, there have been other studies that have reported no correlations between Vitamin E derivatives and either antioxidant activity or carotenoid content, such as Choi et al. [89], who analyzed the grain of rice, sorghum, barley, foxtail millet, and proso millet.

Antioxidant activity and carotenoid content were here significantly and negatively correlated to seed weight. This was also reported by Dong-Yun et al. [90] in an analysis of winter wheat grain. They suggested that seed weight can, therefore, be used indirectly as an index to select varieties with high antioxidant activity, by selecting genotypes that show smaller grain weight. Further studies are needed to better understand this relationship; however, the significant negative and positive correlations for embryo weight with both seed weight and antioxidant activity, along with the significant negative correlation between cotyledon weight and antioxidant activity, suggest that the human selection that was aimed at larger seeds led to an increase in the cotyledon, and in a decrease in the embryo part of the seed. Support for this explanation comes from findings of Kim et al. [91] who analyzed antioxidant activity in soybean varieties and showed that antioxidant activity in all of these was lower for the cotyledon compared to the seed coat and embryo.

### 4.5. Protein

Recently, there has been an ever-increasing demand by consumers for legumes as sources of protein alternative to meat. This interest arises not only because food legume crops represent valuable resources for diet change, but also because of the relative reduction in environmental impact of such protein production [92]. In these accessions, the protein content ranged from 19.3% to 24.8%, in agreement with previous studies that have reported protein contents in different common bean genotypes of ~20% to 30% [9,26,27,30,65]. Statistically significant differences were detected here among these 25 accessions, with the Greek landrace ECe092 showing the highest protein content. However, we detected low to intermediate heritability for this trait (30.4%). Seed protein content heritability is very variable across the different studies that have been carried out, and significant environmental effects have been reported for most grain legumes (for review, see Burstin et al. [93]).

### 4.6. Minerals

Legumes are good sources of essential minerals for human health [94,95], and among legume crops, common bean shows the highest mineral content [9]. Many studies have been performed to characterize the minerals contents of common bean seeds, most of which have been focused on investigations of Fe and Zn [22,24,26,27,30,96,97,98,99,100]. The major interest in Fe and Zn is related to their crucial roles in human health [101,102]. Compared to rice, common bean seeds have 4–10 times the Fe content and 2–3 times the Zn content [103] making common beans an important source of these elements when included in human diet.

Iron is an essential element for human health. It is involved in important metabolic and cellular processes such as deoxyribonucleic acid (DNA) and ribonucleic acid (RNA) synthesis and it is required for synthesis of proteins involved in the oxygen transport, hemoglobin and myoglobin, and other iron-containing enzymes involved in electron transfer (for review see Lieu et al. [104]). High variability for Fe content among the 25 accessions in the present study was found, with maximum of 176.09 mg/kg for the European accession ECe122. Such level of Fe in common bean seeds is among the highest contents that have been reported in the literature. Germplasm screening for the common bean core collection of the International Center for Tropical Agriculture (CIAT) that includes about 1100 genotypes from both gene pools also showed high variability for Fe content, which ranged from 35 mg/kg to 92 mg/kg, with a mean of 55 mg/kg [23,96]. Similar values were reported by Blair et al. [24], Pinheiro et al. [26], and Celmeli et al. [30]. The highest Fe contents were those of Paredes et al. [7] in the Chilean core collection of 246 accessions, where Fe content was up to 152.4 mg/kg. Similar high Fe contents were reported by Philipo et al. [105] in a collection of bean genotypes grown in Tanzania, at 150.8 mg/kg, and by Guzmán-Maldonado et al. [22] in wild and weedy common beans, at 280 mg/kg.

Zinc is the second most important trace element for human after iron as it is involved in many metabolic processes [106]. It is a structural element for transcription factors and acts as cofactor for more than 300 enzymes [107]. Severe Zn deficiency may be involved in the disruption of many organic systems including immune, gastrointestinal, skeletal, and reproductive system [108]. Several studies have focused on Zn content in common bean seeds. Guzmán-Maldonado et al. [22] analyzed wild and weedy common bean genotypes for some essential micronutrients and reported maximum Zn content of 33.1 mg/kg. Celmeli et al. [30] reported a landrace with Zn content of 37.9 mg/kg, with a similar value reported by Brigide et al. [27] for five biofortified cultivars (37.7 mg/kg). Pinheiro et al. [26] analyzed 155 Portuguese landraces and Blair et al. [24] analyzed a wide sample of 365 genotypes (mostly African landraces) for Zn content, and they detected slightly higher values (45.3, 49.1 mg/kg, respectively). Philipo et al. [105] reported a common bean genotype with Zn content of 64.7 mg/kg in an analysis of 99 common bean landraces, cultivars, and breeding lines grown in Tanzania. In the present study, we detected some particularly interesting genotypes (i.e., ECa203, ECe171, ECe095, ECa102) with similarly high Zn contents (46.7, 46.9, 50.4, 50.6 mg/kg, respectively); however, it was the Andean cultivar Midas (ECa046) that showed very high Zn content (77.7 mg/kg), thus making this very interesting for breeding.

However, other macro-elements and micro-elements that are important for human nutrition and health are also present in common bean seeds, such as Ca, Cu, Mg, Mn, K, P, and Se [109]. In the present study, there were significant differences (Tukey tests) among the 25 evaluated accessions for three macro-elements (i.e., Ca, Na, P) and all micro-elements analyzed (i.e., Cu, Fe, Mn, Mo, Ni, Se, Zn). The heritabilities of these elements ranged from 22.4% for P to 84.5% for Na, and thus breeding for development of biofortified varieties for these elements is promising. This applies especially for those showing the highest heritabilities (i.e., Mo, Na, Ni, Se, Zn), although it should be noted that the contents of these elements in seeds can be significantly affected by environmental conditions (e.g., growing site, availability of minerals in the soil, soil type) [22,110].

Sodium showed the highest heritability, and also significant differences among the accessions. Na is needed for humans, as it is involved in several physiological functions; however, it is well documented that the Na consumption greatly exceeds the minimum daily required intake. Such excess of Na in the human diet can have several negative consequences on health (e.g., increased blood pressure), and thus beneficial effects can be obtained by reducing Na intake (for review, see Aburto et al. [111]). Interestingly, for three American accessions, Na was not detected in the seeds (i.e., ECa095, ECa078, ECa060); moreover, four further accessions showed very low Na (American, ECa064, ECa103; European, ECe177, ECe240), thus making them an interesting alternative to be incorporated in low-sodium human diets.

Calcium is a key nutrient for humans, as it is essential for many functions around human health, such as for bone health as it provides rigidity to the skeleton and calcium ions play a role in many metabolic processes. To reach proper Ca intake levels is essential to prevent skeleton system diseases such as osteoporosis and to reduce the risk of bones fractures [112]. Ca intake is usually associate with the consumption of dairy products such as milk, yogurt, and cheese especially in developed countries. Nevertheless, some Asian countries have higher proportion of total Ca intake from non-animal foods such as legumes and grains than from dairy products [113,114]. In this study, wide variability was detected across the present accessions, with European accessions ECe122, ECe177, and ECe166 showing the highest Ca content.

Among the micro-elements, very high heritability was seen for Se content, along with wide variability. Of interest here, for 16 of the 25 accessions no Se was detected, while the remaining accessions showed high Se content. Accessions ECa096 (landrace from Costa Rica) and ECe122 (cultivar from Germany) showed the highest Se contents (4.12, 3.57 mg/kg, respectively). Four other accessions were characterized by high Se content (i.e., ECe294, ECa095, ECe268, ECa203). Celmeli et al. [30] reported lower Se content for 15 common bean landraces and cultivars, with the highest Se content of 0.48 mg/kg. Despite its low levels in the body, Se is essential for human health due to its broad spectrum of biological functions. Se is the principal component of selenoproteins, that are involved in many important enzymatic functions such as production and regulation of levels of thyroid hormone or reduction of nucleotides for DNA synthesis [115]. Se is needed for the regulation of the immune system [115,116]. It has been described the influence of Se in cardiovascular disease risk and in reducing cancer risk [115]. Plants are the main dietary source of Se entering in the food chain through plant-based foods. Considering this, the high heritability observed for Se content in common bean seeds in this study is important for future biofortifying breeding programs aimed to develop Se-enriched plant-based foods that can help in reducing Se related deficiency disorders.

The other micro-elements of Cu, Mn, Mo and Ni showed intermediate to high heritabilities and significant differences among the materials analyzed. The highest contents of each of these four microelements were seen respectively for accessions ECe092, ECa052, ECe234, and ECa095. Determination of the contents of such micro-elements is largely lacking in the literature. The highest Cu content of 16.5 mg/kg and Mn content of 34.8 mg/kg in the present study are similar to the contents reported by Brigide et al. [27] and Pinheiro et al. [26], who indicated maximum Cu contents of 11.7 mg/kg and 13.5 mg/kg, respectively, and Mn contents of 17.91 mg/kg and 20.1 mg/kg, respectively.

Several significant correlations were defined here for the nutritional elements. Considering those with significance level < 0.001, the positive correlations that involved the macro-elements and micro-elements were between: Mg and K; Mn and K; Mn and Ca and Fe and Mn (as also reported by Pinheiro et al. [26]); Mo and Mg; Fe and Na; Fe and Ca; and Ca and Na. Considering also other traits, significant positive correlations were seen for seed weight with Na and with Zn, both embryo weight and antioxidant activity were negatively correlated with Na, carotenoid content was positively correlated with Mn, and Cu with δ-tocopherol. We did not find any significant positive correlations for Fe and Zn, as observed in several other studies [26,30,94].The lack of correlation between these two micro-elements could be explained by the small size of our sample or, considering that we included European individuals showing inter-gene pool introgression, this can have affected the level of Fe and Zn in seeds of our common bean set of materials; in this regard, Blair et al. [24], analyzing a sample of 365 accessions, mostly represented by African landraces, found that the Andean beans are characterized, on average, by a higher Fe and lower Zn content compared to Mesoamerican ones; interestingly they observed that introgressed materials presented the highest Fe content and a level of Zn comparable to Mesoamerican accessions. Correlations between different nutritional compounds can help breeders in improve the quality of seeds by selecting more than one element at a time.

### 4.7. Changes in Chemical Composition of Common Bean Seeds in Europe

The data obtained in the present study can also be seen from another point of view: i.e., as the evolution of the common bean. One of the main evolutionary processes for common bean was its introduction, spread, and adaptation to European agro-ecosystems.

Even if our sample is too small to have a complete picture of the effects of such process in the quality traits of common bean seeds, our findings (both the PCA and ANCOVA) represent a first observation that these events strongly affected not only common bean genetic diversity [20,21], but also its phenotypic diversity at the level of the chemical composition of the seeds.

Principal component analysis showed that the European accessions were more widespread compared to the American accessions, which suggests a higher diversity for seed morphology and chemical composition in the European common bean. This major diversity might have arisen in Europe because the frequency of hybrids between the Mesoamerican and Andean gene pool was more than four-fold that in the Americas, which appears to be due to breakdown of geographic barriers and co-cultivation of the Mesoamerican and Andean forms in close sympatry by farmers [20,21]. The introgression between the gene pools once out of the Americas might have created genotypes that carried new combinations of alleles that resulted in differences in the seed chemical compositions. The combination of PCA with ANCOVA allowed it to be highlighted that the European genotypes have significantly higher seed weights (as reported by Angioi et al. [20], and δ-tocopherol and protein contents, compared to the American genotypes). In contrast, the American genotypes have significantly higher antioxidant activities and carotenoids contents. For the minerals, the European beans had significantly higher Na and Ca contents, while the American beans had significantly higher Mg and Se contents. Even if further studies based on a wider sample of American and European genotypes are needed, these findings have important implications for breeding, inasmuch as they can guide breeders in the choice of the genetic resources to be investigated and exploited on the basis of the traits that they want to improve.

## 5. Conclusions

The characterization of plant genetic resources is crucial for breeders, so that they can exploit the diversity to implement appropriate breeding programs. This was the main aim of the present study, where 25 common bean genotypes were analyzed for nutritional traits.

One of the main outcomes of the present study is the high variability across these samples for almost all of the traits, and the identification of some very interesting genotypes when considering multiple traits. The Greek landrace ECe092 was characterized by the highest lipid, δ-tocopherol, protein, and Cu contents and very high γ-tocopherol, P, and Ni contents. The Spanish landrace ECe312 showed high γ-tocopherol, δ-tocopherol, protein, K, and Cu contents. The Mexican cultivar ECa095 showed high antioxidant activity and carotenoids, Ca, Mn, and Se contents, the highest content of Ni and low Na content. Focusing on the minerals, three accessions are particularly interesting for future breeding: (i) the Spanish landrace ECe234 showed the highest K and Mo contents and high contents of Mg, P, Fe, and Mn were observed for this accession too; (ii) the German cultivar ECe122 showed the highest Ca and Fe content and was among the highest for Mn, Se, and Mo contents; and (iii) the Italian landrace ECe177 was characterized by the highest P content and particularly high Ca and Mo contents and low Na content.

The use of pure lines (i.e., derived by single seed descent) and the evaluation of seeds obtained in a designed field trial with replicates allowed estimation of the heritabilities of the different nutritional traits. This parameter is important, inasmuch as it provides information about the breeding value of the sample. Not all of the traits showed intermediate to high heritability estimates, which indicates a relevant influence of environmental conditions for these nutritional compounds. This suggests the need to conduct more studies on the influence of the environment (e.g., weather, soil) on the expression of common bean nutritional traits, to identify the best agro-ecological conditions that can maximize the nutritional quality of the promising genotypes.

Finally, this study points out the importance to take into account the evolutionary history of crops; indeed, even if more studies are needed in this regard, our findings suggest significant effects on diverse quality traits related to adaptation of the common bean in Europe, out of its centers of origin. Acquiring more knowledge of this evolutionary process can indeed guide geneticists and breeders in their choice of samples to be characterized for different traits.

## Figures and Tables

**Figure 1 foods-10-01572-f001:**
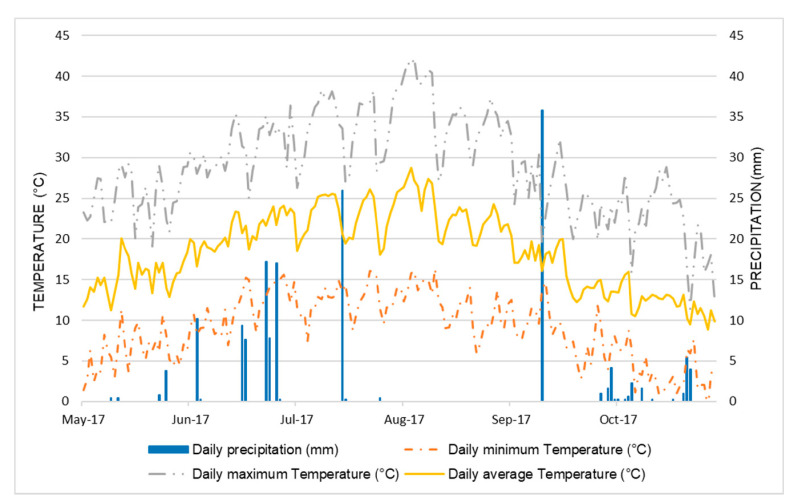
Total precipitation and average temperatures during the bean growing season (May–October 2017).

**Figure 2 foods-10-01572-f002:**
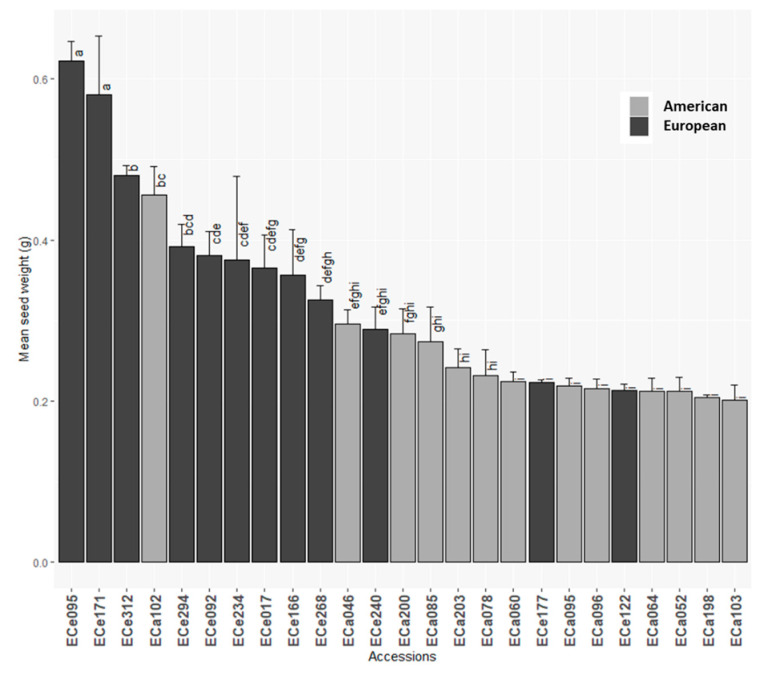
Mean seed weights for the 25 American and European accessions. Columns with different letters are significantly different (*p* < 0.05; Tukey test).

**Figure 3 foods-10-01572-f003:**
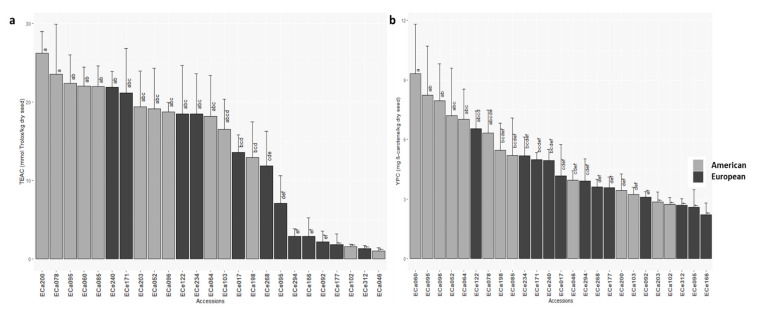
Mean antioxidant activity (TEAC) (**a**) and carotenoids content (YPC); (**b**) for the 25 American and European accessions. Columns with different letters are significantly different (*p* < 0.05; Tukey tests). Ç.

**Figure 4 foods-10-01572-f004:**
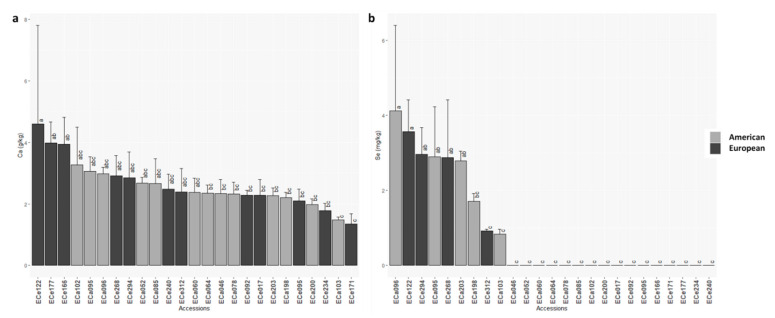
Mean Ca (**a**) and Sel (**b**) contents for the 25 American and European accessions. Columns with different letters are significantly different (*p* < 0.05; Tukey tests).

**Figure 5 foods-10-01572-f005:**
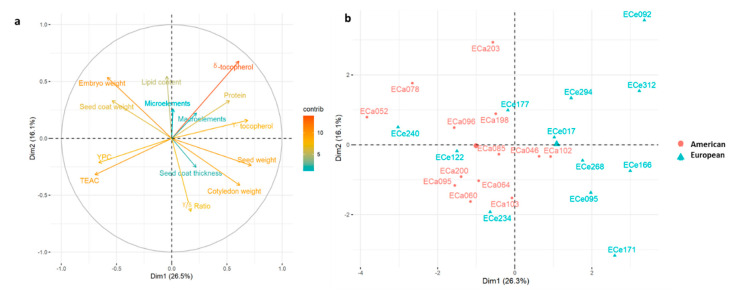
(**a**) Principal component analysis for the 25 American and European accessions based on morphological seed traits (seed, cotyledon, embryo, coat weights), nutritional quality traits (lipid, γ-tocopherol, δ-tocopherol, total tocopherols, YPC, protein contents; TEAC), and macro-element and micro-element contents (sums of Fe, Zn, Mn, Cu, Se, Mo, Ni contents and Na, Mg, P, K, Ca contents); (**b**) loading plot from the principal component analysis. Positive correlated variables point to the same side of the plot, negative to opposite sides. Color gradient (from red to blue) indicates contribution of each variable to each dimension. A, American accessions; E, European accessions.

**Figure 6 foods-10-01572-f006:**
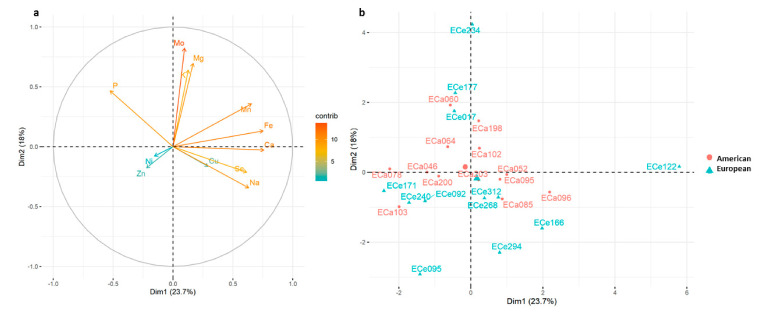
(**a**) Principal component analysis for the 25 American and European accessions based on macro-elements (Ca, K, Mg, Na, P) contents and micro-elements (Cu, Fe, Mn, Mo, Ni, Se, Zn) contents; (**b**) loading plot from the principal component analysis. Positive correlated variables point to the same side of the plot, negative to opposite sides. Color gradient (from red to blue) indicates contribution of each variable to each dimension. A, American accessions; E, European accessions.

**Figure 7 foods-10-01572-f007:**
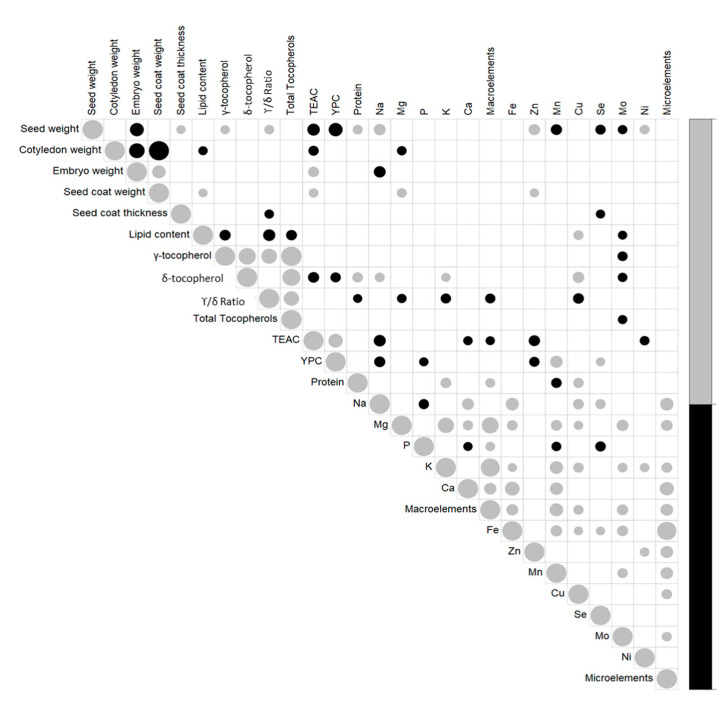
Correlation plot matrix between the traits for the 25 American and European accessions. Only significant correlations are shown (*p* < 0.05). The size of the symbols indicates the value of each correlation. Black, negative correlations; gray, positive correlations (for details, see Appendix A).

**Table 1 foods-10-01572-t001:** Common bean accessions used in the present study.

BEAN_ADAPT Accession Code	Synonym	AM/EU ^1^	Gene Pool ^2^	Country	Biological Status	Growth Habit ^3^	Phaseolin Type	Seed Color	q-Value ^4^
ECa046	MIDAS	AM	And	Argentina	Landrace	I	T	White	0.33
ECa052	G5191	AM	MA	Venezuela	Landrace	II	S	Black	1.00
ECa060	PI196927	AM	MA	Mexico	Landrace	III	S	Black	1.00
ECa064	PI268110	AM	MA	Mexico	Cultivar	V	S	Black	1.00
ECa078	PI309831	AM	MA	Costa Rica	Landrace	III	S	Purple	0.93
ECa085	PI311794	AM	MA	El Salvador	Landrace	V	S	Dark brown	1.00
ECa095	PI325730	AM	MA	Mexico	Cultivar	II	S	Black	1.00
ECa096	PI345574	AM	MA	Costa Rica	Landrace	I	S	Black	1.00
ECa102	W618757	AM	MA	Argentina	Landrace	IV	-	White	1.00
ECa103	BAT93	AM	MA	Colombia	Cultivar	II	S	Light brown	1.00
ECa198	-	AM	MA	Brazil	Landrace	II	-	Black	1.00
ECa200	BZL-0237	AM	MA	Brazil	Landrace	V	-	Dark brown	1.00
ECa203	-	AM	MA	Brazil	Landrace	V	-	Light brown	1.00
ECe017	PHA12	EU	And	Italy	Landrace	I	T	Brown	0.14
ECe092	PHA285	EU	MA	Greece	Landrace	IV	S	White	0.94
ECe095	PHA3094	EU	MA	Yugoslavia	Landrace	IV	-	Brown	0.93
ECe122	PHA6025	EU	MA	Germany	Cultivar	II	-	Black	1.00
ECe166	PHA7094	EU	MA	Bulgaria	Landrace	III	S	White	1.00
ECe171	PHA3524	EU	And	Georgia	Landrace	IV	C	Black	0.00
ECe177	-	EU	MA	Italy	Landrace	III	-	White	0.68
ECe234	PHA7527	EU	And	Spain	Landrace	I	-	Purple, mottled cream	0.09
ECe240	PHA12572	EU	And	Germany	Landrace	I	-	Brown	0.04
ECe268	PHA529	EU	MA	Bulgaria	Landrace	III	-	Light brown, striped brown	1.00
ECe294	PHA6020	EU	MA	The Netherlands	Cultivar	V	-	White	1.00
ECe312	PHA7134	EU	MA	Spain	Landrace	IV	-	White	0.78

^1^ AM, American accessions; EU, European accessions; ^2^ And, Andean gene pool; MA, Mesoamerican gene pool; ^3^ growth habit Type: I, determinate bush; II, indeterminate bush; III, indeterminate prostate; IV, indeterminate climbing; V, determinate climbing; ^4^ membership coefficient to the genetic cluster representing the Mesoamerican gene pool.

**Table 2 foods-10-01572-t002:** ANOVA for the morphological and compositional traits analyzed.

Trait	Units	Minimum	Maximum	Estimated h^2^ (%)	Significance
Seed weight	g	0.2	0.6	91.4	***
Cotyledon weight	mg/g seeds	879.1	924.3	36.6	***
Embryo weight	mg/g seeds	9.1	16.1	52.7	***
Seed coat weight	mg/g seeds	62.7	106.2	39.6	***
Seed coat thickness	mm	0.1	0.2	10.2	ns
Lipid content	g/100 g dry seed	1.8	3.1	0	ns
γ-Tocopherol	mg/100 g of lipids	43.3	127.2	12.7	ns
δ-Tocopherol	mg/100 g of lipids	3.7	13.6	31.5	***
γ/δ ratio	-	7.7	19.8	24.0	ns
Total tocopherols	mg/100 g of lipids	47.0	134.0	10.2	ns
TEAC	mmol Trolox/kg dry seed	1.1	26.2	84.1	***
YPC	mg β-carotene/kg dry seed	2.2	9.3	69.4	***
Protein	%	19.3	24.8	30.5	***
Na	g/kg	0.0	0.5	84. 5	***
Mg	g/kg	1.8	2.4	16.2	ns
P	g/kg	3.4	5.7	22.4	*
Ca	g/kg	1.4	4.6	35.9	***
K	g/kg	16.3	20.8	7.8	ns
Total macro-elements	g/kg	24.6	30.4	12.5	ns
Cu	mg/kg	7.0	16.5	35.3	***
Fe	mg/kg	55.4	176.1	39.5	***
Mn	mg/kg	15.9	34.8	38.9	***
Mo	mg/kg	0.7	4.3	67.3	***
Ni	mg/kg	0.0	4.9	52.2	***
Se	mg/kg	0.0	4.1	81.2	***
Zn	mg/kg	5.1	77.7	53.3	***
Total micro-elements	mg/kg	106.1	252.8	38.6	***

TEAC, trolox equivalent antioxidant activity; YPC, yellow pigment content. * *p* < 0.05; *** *p* < 0.001; ns: not significant (ANOVA).

**Table 3 foods-10-01572-t003:** Analysis of covariance for the mean morphological seed traits for the American and European accessions.

Accession	Seed Weight (g)	Cotyledon Weight (mg/g)	Embryo Weight (mg/g)	Seed Coat Weight (mg/g)	Seed Coat Thickness (mm)
American	0.25	904.6	13.28	82.06	0.12
European	0.38	908.7	11.83	79.53	0.13
Significance	***	ns	*	ns	ns

* *p* < 0.05; *** *p* < 0.001; ns: not significant.

**Table 4 foods-10-01572-t004:** Analysis of covariance for the mean nutritional quality traits for the American and European accessions.

Accession	Lipids (g/100 g Dry Seed)	γ-Tocopherol (mg/100 g of Lipids)	δ-Tocopherol (mg/100 g of Lipids)	γ/δ Ratio	Total Tocopherols (mg/100 g of Lipids)	TEAC (mmol Trolox/kg Dry Seed)	YPC (mg β-Carotene/kg Dry Seed)	Protein (%)
American	2.41	75.70	6.60	11.47	82.29	17.20	5.61	21.02
European	2.30	87.02	7.77	11.37	92.82	10.30	3.97	21.78
Significance	ns	ns	**	ns	ns	***	***	*

TEAC, trolox equivalent antioxidant activity; YPC, yellow pigment content. * *p* < 0.05; ** *p* < 0.01; *** *p* < 0.001; ns: not significant.

**Table 5 foods-10-01572-t005:** Analysis of covariance for the mean micro-elements and macro-elements for the American and European accessions.

Accession	Macro-Elements (g/kg) and Micro-Elements (mg/kg)													
	Na	Mg	P	K	Ca	Macro-Elements	Fe	Zn	Mn	Cu	Se	Mo	Ni	Micro-Elements
American	0.113	2.112	4.250	18.427	2.427	27.364	78.56	33.59	25.30	11.19	0.95	1.81	2.76	154.16
European	0.233	1.995	4.673	18.423	2.751	28.074	88.70	31.71	23.92	11.83	0.86	1.97	2.78	159.92
Significance	***	**	ns	ns	**	ns	ns	ns	ns	ns	***	ns	ns	ns

** *p* < 0.01; *** *p* < 0.001; ns: not significant.

**Table 6 foods-10-01572-t006:** Correlation coefficients between the principal components, PC1 and PC2, and seed morphological and compositional traits. Only significant correlations are shown (*p* < 0.05).

Principal Component	Trait	Pearson’s Coefficient (r)	*p*-Value
PC1	Seed weight	0.72	<0.001
	ϒ-tocopherol	0.68	<0.001
	Cotyledon weight	0.61	0.001
	δ -tocopherol	0.61	0.001
	Protein	0.52	0.007
	Seed coat weight	−0.53	0.005
	Embryo weight	−0.58	0.002
	YPC	−0.66	<0.001
	TEAC	−0.69	<0.001
PC2	δ -tocopherol	0.68	<0.001
	Lipid content	0.55	0.005
	Embryo weight	0.54	0.006
	Cotyledon weight	−0.41	0.040
	ϒ/δ Ratio	−0.64	0.001

**Table 7 foods-10-01572-t007:** Correlation coefficients between the principal components, PC1 and PC2, and macro-elements (Na, Mg, P, K, Ca) contents and micro-elements (Fe, Zn, Mn, Cu, Se, Mo, Ni) contents. Only significant correlations are shown (*p* < 0.05).

Principal Component	Trait	Pearson’s Coefficient (r)	*p*-Value
PC1	Ca	0.76	<0.001
	Fe	0.75	<0.001
	Mn	0.66	<0.001
	Na	0.63	0.001
	Se	0.62	0.001
	P	-0.53	0.007
PC2	Mo	0.82	<0.001
	Mg	0.70	<0.001
	K	0.64	0.001
	P	0.47	0.019

## Data Availability

Data is contained within the article and/or Appendix A.

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
