# Peer review of "Characterization of Nutritional Quality Traits of a Common Bean Germplasm Collection"

_foods, 2021, doi:10.3390/foods10071572_

Round 1
Reviewer 1 Report
The manuscript reports data on the evaluation of a number of seed morphological and nutritional traits of 25 single seed descent common bean domesticated accessions characterized by a high genetic diversity and originating from America (Mesoamerican and Andean materials) and Europe (materials in which intergene-pool hybridization occurred). The nutritional traits considered were proteins, lipids, γ- and δ-tocopherols, antioxidant activity, carotenoids content and macro- and micro-elements composition. The study is well described and results underwent a comprehensive statistical analysis. I appreciated the evolutionary approach to underline and suggest that introgression between gene pools occurred during common bean adaptation in Europe has been accompanied by combination of alleles resulting in differences in nutrients composition.
I have only some minor remarks:
Lines 27-28: it is not clear which findings support the link between the variability of the observed traits with the effects of environmental growth conditions on seed compositional traits.
Lines 88-82: it is stressed the importance of considering materials grown in different specific field trials involving replicates as quality traits are influenced by environmental conditions.
In both cases, it should be better explained the importance of sampling criteria (as clearly explained in the discussion, lines 492-494) here and in M&M section.
Discussion
Lipids and tocopherols: considering the low content of lipids, what is the impact of the observed (significant) differences in γ-tocopherols on the RDA? What is the real potential for human health?
Antioxidant activity and carotenoid content: considering the importance (and possible abundance) of polyphenols for the antioxidant activity, the quantification of this class of compounds should have been included in the analyses. Without this analysis it is very speculative try to address a significant role of tocopherol and/or carotenoids in the observed antioxidant activities.
Lines 752-753: several papers reported the existence of a correlation between Fe and Zn content, however you did not observe it, could you provide an explanation for this? Could it be due to the dimension of the sample?
Reviewer 2 Report
The Authors of the manuscript FOODS-1271776 entitled “Characterization of nutritional quality traits of a common bean germplasm collection” focused on investigation the detailed chemical composition of 25 genetically different purified accessions of Phaseolus vulgaris L. The Authors obtained accessions of both European and American origin and performed their characterization in terms of morphology and nutrients (protein, carotenoids, tocopherols and minerals) content as well as antioxidant properties. They have decided to use wide range of statistical methods to analyze their results which raises the value of their research. I have found this paper very well planned, performed and presented and there are few small inaccuracies, which should be addressed and corrected before publication. You may find those detailed remarks listed below:
- Lines 219-221: Detailed parameters of the mineralization process (time, temperature, pressure) should be presented.
- In many cases number of decimal places is to big (e.g. 252.75). It is enough to present the results with four significant digits, so the Authors are asked to check the whole manuscript, tables, figures and supplementary materials for this.
- All figures and supplementary figures: error bars representing standard deviation are needed.
- Line 281: 1031.2mg/g - how it is possible? 1031.2 mg = 1.0312 g, so how it can be 1.0312g/1g? Please explain. The Authors should carefully checked all the results.
- Moreover, the units for many results are sometimes confusing, e.g. mg/kg ß-carotene (line 301), 4.606 g/kg (line 313 – kg of what?), or in Table S2 mg/100 oil? The Authors should clearly state how the results are expressed.
- Line 310: “ANOVA was significant” it is a jargon term and it should be replaced
Reviewer 3 Report
This proposed manuscript intended to characterize and describe nutritional quality traits of different American and European domestic genotypes of common bean (purified accessions) to identify potential sources of improved seed quality traits that can be used in future breeding programmes. The authors found a high variability across these samples for almost all the traits and also they identified very interesting genotypes when considering multiple straits.
Concerning to the quality of the paper: The title of this paper clearly reflects its content. The objectives are clear and appropriate in the view of this subject matter. The results and discussion of this study is well written. In general, references are recent and adequate.
It is my oppinion that figures must be improved in quality printing.
Author Response
Please see the attachment.

This manuscript is a resubmission of an earlier submission. The following is a list of the peer review reports and author responses from that submission.
Round 1
Reviewer 1 Report
The work by Murube et al., aims at describing the nutritional traits of 25 common bean accessions (landraces and cultivars). Note that the study was done on plants that were grown on the same place at the same time in a random block assay.
The study provides interesting results on nutritional values of the bean seeds and also some information related to cultivated bean evolution. This study represents a great amount of work and provides a lot of data valuable for nutritionist and to bean breeders.
The work is of broad interest but also a great source of information for bean breeders. The paper is well written and data well presented. The discussion is synthetizing nicely the data.
For me this work can be published as it stands.
Reviewer 2 Report
In this ms the authors report the morphological and compositional quality traits of 25 American and European common bean seeds. Significant differences were identified among them that are described and analysed in grait detail. These are the results.
In the abstract the authors indicate “the importance of considering the effects of environmental growth conditions on seed compositional traits” but they don’t report results on this interesting aim.
In addition, in the abstract they comment on “the adaptation following the introduction and spread of common bean in Europe that seems to have affected the nutritional profile of the seeds”, but these are initial observations that need more data.
In conclusion, considering the data reported in this ms, I would suggest the authors to send it to an applied agricultural research journal.
Reviewer 3 Report
In the present manuscript, seeds of 25 American and European common bean purified accessions were evaluated for different morphological and compositional quality traits.
The use of single rows (9 seeds per row) per plot does not allow to use some plants as outliers. Therefore, the measurements are limited to the plants of a single row.
What is the growing period for common bean in the studied area?
What was the fertilization regime? What were the soil conditions?
Provide meteorological data.
More details are needed about sampling. When harvesting of pods took place? Where all the accessions harvesetd at the same time?
The authors should have measured phytates or phytic acid which is an antinutritional factor commonly found in common bean.
Round 2
Reviewer 3 Report
The authors have answered to my comments. Therefore, I recommend the acceptance of the manuscript in its present form.